# Fast and Scalable Method for Efficient Multimodal Feature Extraction with Optimized Maximal Correlation

## Abstract

This paper introduces the UniFast HGR framework, a novel method designed to enhance the computation of Hirschfeld-Gebelein-Rényi (HGR) maximal correlation, specifically optimized for large-scale neural networks and multimodal tasks. UniFast HGR introduces a variance constraint and optimizes the trace term, resulting in a more accurate approximation of the original HGR. By replacing traditional covariance-based measures with cosine similarity and eliminating bias from the main diagonal, the approach significantly reduces computational complexity while enhancing overall accuracy. These improvements make UniFast HGR highly scalable and capable of delivering superior performance in diverse, large-scale multimodal learning applications. Building on this foundation, the OptFast HGR method further optimizes performance by reducing the number of normalization steps, achieving efficiency and computational cost comparable to dot product and cosine similarity operations. This advancement accelerates computation without sacrificing performance. Experimental results indicate that UniFast HGR effectively balances efficiency and precision, establishing it as a robust solution for modern deep learning challenges.

## 1 Introduction

In machine learning, the extraction of informative and generalizable data representations is critical (Bengio et al., 2013). This task becomes increasingly complex when working with multimodal data, which encompasses information from diverse sources such as images, text, and audio (Summaira et al., 2021). Human cognition inherently integrates these disparate data types, facilitating more accurate interpretation and decision-making. However, machines encounter substantial difficulties in synthesizing such heterogeneous information, primarily due to the distinct statistical properties inherent in each modality. These differences obscure the correlations that are vital for learning effective feature representations (Baltrusaitis et al., 2018; Guo et al., 2019; Gandhi et al., 2023). Traditional methods, such as Canonical Correlation Analysis (CCA)(Hotelling, 1936) , have been employed to identify linear relationships between two datasets, while other approaches, such as minimizing Euclidean distances between feature spaces, have also been explored (Frome et al., 2013).

The Hirschfeld-Gebelein-Rényi (HGR) maximal correlation (Hirschfeld, 1935; Gebelein, 1941; Rényi, 1959) has been widely recognized as a robust metric for capturing nonlinear dependencies between random variables. Its application in machine learning, particularly for multimodal data integration, has garnered attention due to its theoretical ability to extract maximally informative features across modalities (Huang et al., 2017). Despite its potential, the practical implementation of HGR maximal correlation in modern machine learning frameworks presents significant challenges.

The original HGR maximal correlation framework imposes strict whitening constraints, necessitating uncorrelated feature representations. This requirement introduces substantial computational burdens, especially when processing high-dimensional data common in deep neural networks. Matrix inversion and decomposition operations, required for whitening, are computationally expensive and susceptible to numerical instability, thus limiting their scalability in large-scale machine learning applications. Efforts to overcome these limitations have led to the development of extensions

such as Kernel CCA (Akaho, 2006) and Deep CCA (Andrew et al., 2013), which aim to approximate HGR maximal correlation. However, these methods remain constrained by their transformation functions and continue to suffer from computational inefficiencies stemming from whitening. Alternative approaches, such as Soft-CCA and Correlational Neural Networks, attempt to alleviate these constraints but risk altering the underlying feature geometry, which can reduce the discriminative capacity of the extracted features (Chang et al., 2018; Chandar et al., 2016). A further limitation of the HGR maximal correlation framework is its lack of optimization for supervised learning tasks. The method presumes that discriminative information is inherently preserved within the shared subspace of different modalities. In practice, however, this assumption often fails, especially in scenarios where modalities are weakly correlated or contain substantial modality-specific information. Maximal Correlation Regression (MCR) addresses this issue by incorporating HGR maximal correlation to derive analytically optimal weights for supervised learning, demonstrating strong theoretical connections to established methods such as linear discriminant analysis and softmax regression. MCR has been shown to achieve competitive performance on various real-world datasets (Xu & Huang, 2020). Additionally, recent research has explored the sample complexity involved in estimating HGR maximal correlation functions using the Alternating Conditional Expectations (ACE) algorithm. This work provides error bounds and identifies optimal sampling strategies for large datasets in both supervised and semi-supervised learning contexts (Huang & Xu, 2021). In the domain of multimodal fusion, HGR maximal correlation has also been successfully incorporated into loss functions to enhance person recognition performance across multimodal data sources (Liang et al., 2021).

To address these limitations, the Soft-HGR framework (Wang et al., 2019) was introduced, providing a more flexible alternative by relaxing the whitening constraints while preserving the essential geometry of the feature space. This framework utilizes a low-rank approximation based on the empirical distribution of the dataset, which can be extended to accommodate missing modalities and incorporate supervised information. A deep learning framework has also been developed to address challenges in audio-visual emotion recognition, such as missing labels and incomplete modalities, by employing an HGR maximal correlation-based loss function to unify and capture essential information from diverse training data (Ma et al., 2021). Additionally, a multimodal conditional GAN has been introduced as an efficient data augmentation method for audio-visual emotion recognition, although this approach modifies the transmitted data during the fusion process (Ma et al., 2022). In the MultiEMO study, Soft-HGR was applied to correlation analysis, leading to enhanced classification accuracy in emotion recognition (Shi & Huang, 2023). Despite these advancements, Soft-HGR continues to face challenges when applied to complex neural architectures and large-scale datasets. Its scalability and efficiency, while improved, remain insufficient to meet the demands of modern deep learning applications.

With the advent of large-scale models and extensive datasets, the limitations of the Soft-HGR framework have become increasingly pronounced. Although Soft-HGR has demonstrated utility in certain applications, its computational complexity and inefficiency present significant obstacles for its integration into state-of-the-art deep learning architectures, especially when applied to large-scale data and models. The practical utility of HGR maximal correlation, along with its extensions, remains hindered by critical challenges, including excessive computational overhead, significant resource requirements, unstable performance improvements, and insufficient scalability. A more efficient and scalable solution is urgently required to fully exploit the potential of multimodal learning in contemporary machine learning environments.

To address the limitations of previous frameworks, UniFast HGR is introduced as an advanced solution designed to overcome computational bottlenecks and scalability challenges. UniFast HGR features an optimized algorithmic structure that reduces computational overhead, improves discriminative accuracy, and offers a unified approach scalable to large datasets and deep models. Additionally, it is engineered to fully leverage deep neural networks, enabling efficient and scalable learning of correlated features across multiple modalities. The contributions of this framework are as follows:

**Unified Efficiency and Scalability**: UniFast HGR merges the strengths of both traditional HGR and Soft-HGR, addressing their limitations in dimensionality and computational complexity. By integrating the original HGR maximal correlation framework with refinements from Soft-HGR, UniFast HGR achieves stable and precise feature extraction within a bounded range of [-1,1]. This integration enhances adaptability and performance within modern deep learning architectures, making the framework particularly well-suited for large-scale datasets and deep neural networks.

**Enhanced Discriminative and Correlation Power**: UniFast HGR incorporates discriminative objectives, enabling the extraction of highly informative features for downstream supervised tasks. Additionally, it enhances the correlation between data modalities by optimizing the function maximization process, ensuring effective alignment of correlated information. This improvement is particularly important in complex neural architectures, where maintaining strong multimodal correlations directly impacts overall model performance. The framework also substitutes traditional covariance-based correlations with cosine similarity, improving the accuracy of correlation measures. Furthermore, optimizations in the trace term, including the exclusion of diagonal elements, prevent bias introduced by self-correlations, ensuring more accurate results. UniFast HGR strikes a balance between speed and performance, offering an efficient yet powerful solution for diverse deep learning applications. Collectively, these enhancements improve the framework's ability to extract and utilize meaningful correlations, thus increasing both discriminative power and correlation accuracy across various tasks.

**Overcoming Complexity Limitations**: UniFast HGR resolves the complexity and inefficiency issues associated with Soft-HGR, providing a faster, more scalable solution for large-scale deep learning applications. Additionally, the OptFast HGR variant further optimizes performance by reducing the number of normalization steps, achieving computational efficiency comparable to dot product and cosine similarity operations. This optimization significantly accelerates processing while maintaining high performance. These advancements represent a substantial step forward in applying HGR maximal correlation, particularly in managing dimensionality challenges and enabling more effective multimodal learning at scale.

## 2 RELATED WORK

### 2.1 HGR CORRELATION ANALYSIS AND LIMITATIONS

HGR maximal correlation extends Pearson correlation by providing a more comprehensive measure of dependency, originally developed for single features but naturally extendable to multiple features. In the case of random variables $x$ and $y$, which share a joint distribution across the domains $X$ and $Y$. Given $f = [f_1, f_2, \cdots, f_k]^T$ and $g = [g_1, g_2, \cdots, g_k]^T$, the maximal correlation for a set of $k$ features in the HGR framework is defined as follows:

$$\rho^k(X,Y) = \sup_{\substack{f:x \to R^k, \frac{1}{k}[f]=0, Cov(f)=I \\ g:y \to R^k, E[g]=0, Cov(g)=I}} E\left[f^T(X)g(Y)\right] \qquad (1)$$

where $k$ represents the dimension of the data.

The HGR maximal correlation is determined through optimization over sets of Borel measurable functions, which are characterized by zero mean and stable covariance. This correlation, ranging from 0 to 1, signifies either complete independence or a deterministic relationship between $X$ and $Y$. However, the computational complexity of HGR maximal correlation arises primarily from the whitening constraints, which necessitate matrix inversion and decomposition, resulting in a time complexity of $O(K^3)$. These challenges are compounded by scalability issues, particularly as covariance matrices can become ill-conditioned, leading to gradient explosions in high-dimensional spaces.

Soft-HGR builds on the HGR framework by seeking an optimal solution to maximal correlation under specific whitening constraints, while introducing a low-rank approximation to mitigate some of the computational challenges posed by HGR (Wang et al., 2019). This method facilitates integration with neural networks by circumventing the computational difficulties of whitening constraints, enabling efficient computation of maximal correlations. Soft-HGR focuses on extracting highly correlated feature mappings from diverse random modalities without strictly relying on whitening. Applied to mini-batches, Soft-HGR reduces complexity to $O(mK^2)$ by approximating batch covariance, offering improved stability even for large feature dimensions. Despite these advancements, Soft-HGR introduces new challenges during the fusion process, where data values can be modified. Output values from Soft-HGR can become excessively large—sometimes reaching thousands—due to higher network outputs corresponding to higher HGR correlation (Zhang et al., 2024). This sensitivity to signal variance, coupled with large deviations from the ideal HGR, complicates

the comparison of Soft-HGR values across datasets, particularly in cases with a large number of features. As a result, its practical application is hindered. Although low-rank approximation techniques help mitigate some of the computational burden associated with traditional HGR, Soft-HGR still involves more complex operations than simpler alternatives such as dot product. Operations such as covariance matrix calculation, matrix decomposition or inversion, and iterative optimization of feature mappings further contribute to the computational complexity.

These limitations lead to higher computational costs when applying Soft-HGR to large-scale datasets and deep models, complicating its scalability and impeding its efficiency and stability in real-world applications. Consequently, Soft-HGR is less suited for widespread deployment in large-scale deep learning environments. Soft-HGR is mathematically represented as follows:

$$
\max_{f,g} \mathbb{E}\left[f^T(X)g(Y)\right] - \frac{1}{2}\,tr(\mathrm{cov}(f(X))\mathrm{cov}(g(Y)))
$$
$$
\text{s.t. } E[f(X)] = E[g(Y)] = 0
$$
(2)

where $f(X)$ and $g(Y)$ are feature mappings derived from various random modalities.

## 3 PROPOSED METHOD

This section presents the UniFast HGR framework, an advanced solution that significantly improves upon both Soft-HGR and the original HGR maximal correlation approaches. Designed to address computational challenges, scalability limitations, and practical constraints in large-scale neural network applications, UniFast HGR enhances both discriminative and correlation capabilities, facilitating the extraction of highly informative features across diverse data modalities. The following sections outline the key components and innovations of the UniFast HGR framework.

### 3.1 OPTIMIZED CORRELATION FRAMEWORK

#### 3.1.1 VARIANCE CONSTRAINT

To overcome the limitations of Soft-HGR, particularly its sensitivity to changes in signal variance, variance constraints were introduced. Unlike Soft-HGR, which did not enforce variance normalization, the UniFast HGR framework incorporates variance constraints during the optimization process. By definition of HGR maximal correlation, a zero mean and unit variance (Var = 1) are enforced in the Soft-HGR objective, as shown in Formula 2. For the first term of Formula 2, the following holds:

$$
\mathbb{E}\left[f^T(X)g(Y)\right] = \frac{1}{N-1}\sum_{i=1}^{N} f^T(x)g(y)
$$
(3)

By ensuring a mean of zero, the following condition is satisfied:

$$
\mathbb{E}\left[f^T(X)g(Y)\right] = \frac{1}{N-1}\sum_{i=1}^{N} (f(x) - \mathbb{E}[f(x)])^T (g(y) - \mathbb{E}[g(y)])
$$
(4)

where $\mathbb{E}[f(x)]$ and $\mathbb{E}[g(y)]$ represent the means of $f(x)$ and $g(y)$, respectively.

By introducing the variance constraint Var = 1, the following expression is obtained:

$$
E\left[f^T(X)g(Y)\right] = \frac{1}{N-1}\sum_{i=1}^{N} \frac{(f(x) - E[f(x)])(g(y) - E[g(y)])}{\sqrt{\mathrm{Var}[f(x)]}\sqrt{\mathrm{Var}[g(y)]}}
$$
(5)

This variance normalization ensures that the output values of Soft-HGR remain within the range [-1,1]. A key aspect of this method is that as Soft-HGR output values approach 1, the corresponding HGR values also approach 1, due to the synchronous nature of their derivatives (i.e., both rates

of change share the same sign). This correlation allows the use of an HGR approximation under ideal conditions to replace the actual HGR value, improving accuracy while slightly increasing computational complexity. However, by transforming the first term of Equation 2 into a cosine similarity calculation, the computational burden is reduced.

### 3.1.2 Expansion of the Trace Term

The introduction of variance constraints in the Soft-HGR objective increases computational load. However, by expanding the trace term, this additional burden can be mitigated, optimizing the process. The trace term, which plays a critical role in the framework, was not significantly impacted in the original Soft-HGR due to the absence of variance constraints. However, with variance constraints in place, the trace term becomes essential, as it represents the correlation between two matrices or data sets. In refining the Soft-HGR framework, two key components were identified: (1) the correlation between individual elements, and (2) the correlation between the similarity matrices of these sets. Specifically, for a matrix representing the correlation of elements within a set (e.g., set 1 and set 2, as shown in Figure 1), the trace term captures the similarity between the correlation matrices of these sets. This is achieved by expanding the matrix and quantifying the similarity in the distribution of elements. In essence, the trace term provides a more refined measure of the correlation between the sets by capturing the similarity between their respective similarity matrices.

The definition of the trace term is given as follows:

$$trace = \frac{1}{2} \, tr(\text{cov}(f(X), g(Y))) \tag{6}$$

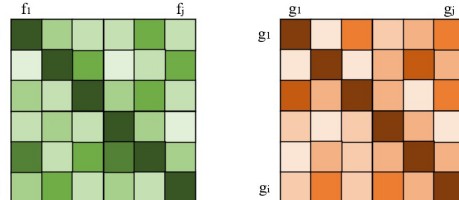

Figure 1: Trace term: Correlation between the similarity matrices of two modalities.

The covariance matrices are computed as follows:

$$\text{cov}[f(X)] = \frac{1}{N-1} \sum_{i=1}^{N} \left( f(x) - \mathbb{E}[f(x)] \right) \left( f(x) - \mathbb{E}[f(x)] \right)^T \tag{7}$$

where $\text{cov}[f(X)]_{ij} = cov[f_i, f_j] \equiv cov f_{ij}$

$$\text{cov}[g(Y)] = \frac{1}{N-1} \sum_{i=1}^{N} \left( g(y) - \mathbb{E}[g(y)] \right) \left( g(y) - \mathbb{E}[g(y)] \right)^T \tag{8}$$

where $\text{cov}[g(Y)]_{ij} = cov[g_i, g_j] \equiv cov g_{ij}$

Considering the trace term,

$$trace = \frac{1}{2} \, tr(\text{cov}(f(X), g(Y))) = \frac{1}{2(N-1)} \sum_{i=1}^{N} \sum_{j=1}^{N} \left( cov f_{ij} - \mathbb{E}[cov f_i] \right) \left( cov g_{ji} - \mathbb{E}[cov g_j] \right) \tag{9}$$

By incorporating the variance constraint Var = 1,

$$trac = \frac{1}{2} \, tr \left( \text{cov}(f(X))\text{cov}(g(Y)) \right) = \frac{1}{2(N-1)} \sum_{i=1}^{N} \sum_{j=1}^{N} \frac{\left( \text{cov} f_{ij} - E\left[ \text{cov} f_i \right] \right) \left( \text{cov} g_{ji} - E\left[ \text{cov} g_j \right] \right)}{\sqrt{\text{Var} \left( \text{cov} f_i \right)} \sqrt{\text{Var} \left( \text{cov} g_j \right)}} \tag{10}$$

Simplifying this expression demonstrates that it is related to the trace of the product of the covariance matrices in the simplified HGR approximation formula. This optimization reduces computational complexity while maintaining the accuracy of the HGR approximation.

## 3.2 UNIFAST HGR

### 3.2.1 SUBSTITUTION WITH COSINE SIMILARITY

In this step, the original covariance-based correlation computations were replaced with cosine similarity to accelerate the calculation process. This substitution was based on the observation that cosine similarity effectively captures relationships between elements while reducing computational complexity, particularly when combined with the expanded trace term. From the definition of cosine similarity, the following holds:

$$\cos(f, g) = \frac{f \cdot g}{\|f\| \|g\|} \tag{11}$$

If all components of a random vector are independent, the square of the vector's modulus will equal the sum of the variances of each component. Thus, Equations 5 and 11 are equivalent:

$$\mathbb{E} \left[ f^T (X) \, g (Y) \right] = \frac{1}{N-1} \sum_{i=1}^{N} \cos(f(X), g(Y)) \tag{12}$$

Similarly, the covariance calculation in Formula (10) can be converted into a cosine similarity calculation:

$$trac = \frac{1}{2} \, tr \left( \text{cov}(f(X))\text{cov}(g(Y)) \right) = \frac{1}{2(N-1)} \sum_{i=1}^{N} \sum_{j=1}^{N} \frac{\left( \cos f_{ij} - E\left[ \cos f_i \right] \right) \left( \cos g_{ji} - E\left[ \cos g_j \right] \right)}{\sqrt{\text{Var} \left( \cos f_i \right)} \sqrt{\text{Var} \left( \cos g_j \right)}} \tag{13}$$

That is,

$$trace = \frac{1}{2} \, tr(\text{cov}(f(X), g(Y))) = \frac{1}{2(N-1)} \sum_{i=1}^{N} \cos(distri\_f, distri\_g) \tag{14}$$

where $distri\_f = f \cdot f^T$ and $distri\_g = g \cdot g^T$

Finally, the UniFast-HGR is computed as follows:

$$\max_{f,g} \mathbb{E} \left[ f^T(X) \right] \mathbb{E} \left[ g(Y) \right] - \frac{1}{2} tr(\text{cov}(f(X), g(Y))) = \frac{1}{N-1} \sum_{i=1}^{N} \cos(f(x), g(y)) - \frac{1}{2(N-1)} \sum_{i=1}^{N} \cos(distri\_f, distri\_g)) \tag{15}$$

That is,

$$UF - HGR = \frac{1}{N-1} \sum_{i=1}^{N} \cos(f(x), g(y)) - \frac{1}{2(N-1)} \sum_{i=1}^{N} \cos(distri\_f, distri\_g)) \tag{16}$$

where $cos(f, g)$ represents the cosine similarity between $f$ and $g$, and $trace(cov(f)cov(g))$ represents the trace of the product of the covariance matrices of $f$ and $g$.

The calculation process for the proposed UF-HGR algorithm is detailed in Algorithm 1.

---

**Algorithm 1** UniFast HGR algorithm

---

**Input:** $m \times n$ feature matrix of $\boldsymbol{f}, \boldsymbol{g}$

**Output:** Objective value of UniFast HGR

1. Normalization:
   $\mathbf{f} \leftarrow \frac{f}{\|f\|}, \mathbf{g} \leftarrow \frac{g}{\|g\|}$
2. Calculation of the cosine correlation coefficient between $f$ and $g$:
   $\cos(f,g) = f \cdot g$
   $corr = \frac{1}{N-1} \sum_{i=1}^{N} \cos(f,g) = \frac{1}{N-1} \sum_{i=1}^{N} f \cdot g$
3. Calculation of the distribution matrix:
   $distri\_f = f \cdot f^T, \quad distri\_g = g \cdot g^T$
4. Initialization processing:
   $distri\_f \leftarrow$ The upper triangular part of $distri\_f$ is extracted using the torch.triu function, excluding diagonal elements
   $distri\_g \leftarrow$ The upper triangular part of $distri\_g$ is extracted using the torch.triu function, excluding diagonal elements
   The symmetry of $distri\_f$ and $distri\_g$ is utilized to restore the upper triangular part to complete matrix.
5. Normalization:
   $distri\_f \leftarrow \frac{distri\_f}{\|distri\_f\|}, \quad distri\_g \leftarrow \frac{distri\_g}{\|distri\_g\|}$
6. Calculation of the cosine correlation coefficient between $distri_f$ and $distri_g$:
   $tr = \frac{1}{N-1} \sum_{i=1}^{N} \cos(distri\_f, distri\_g) = \frac{1}{N-1} \sum_{i=1}^{N} distri\_f \cdot distri\_g$
7. Calculation of the UniFast HGR objective:
   $\frac{1}{N-1} \sum_{i=1}^{N} \cos(f,g) - \frac{1}{2(N-1)} \sum_{i=1}^{N} \cos(distri\_f, distri\_g)$

---

### 3.2.2 REMOVING THE MAIN DIAGONAL

A key enhancement in the development of UniFast HGR involved removing the main diagonal of the correlation matrices. The diagonal entries, inherently 1 due to the variance constraint (Var = 1), represent self-correlations that skew cosine similarity calculations by disproportionately influencing the angle, leading to overestimated similarity. By eliminating the diagonal, this issue is mitigated, as the fixed diagonal value of 1 biases the resulting vector toward a specific angle, narrowing the range of variation and reducing accuracy. Moreover, correlation values, typically between [-1,1], are further distorted by the multiplication effect, which amplifies the diagonal's influence and diminishes the contribution of non-diagonal elements, skewing the similarity measure. This distortion causes the calculated angles to align with the maximum diagonal value, limiting the ability of other values to approach 1 and often pushing them significantly below 1. By removing the diagonal, a more accurate and representative similarity measure is achieved. Optimizing the trace term by eliminating diagonal elements, which correspond to self-correlations, ensures that it reflects the correlation between the two sets more accurately. This enhancement improves both computational efficiency and accuracy, making UniFast HGR not only faster but also more precise, aligning closely with the theoretical expectations of HGR.

### 3.3 GENERALIZATION TO MORE MODALITIES

The HGR maximum correlation was originally defined for two random variables, and extending this correlation-based approach to multiple modalities presents significant challenges. The introduction of additional modalities imposes new whitening constraints, thereby increasing computational complexity. However, UniFast HGR offers enhanced flexibility in managing this complexity. To handle two or more modalities, the multimodal UniFast HGR must be capable of learning and simultaneously recording all paired feature transformations. Assuming that $X_1, X_2, \ldots, X_m$ are $m$ different modalities, and $f(1), f(2), \ldots, f(m)$ denote their corresponding transformation functions. The multimodal UniFast HGR is defined as follows:

$$UF - HGR = \frac{1}{N-1} \sum_{j=k}^{m} \sum_{i=1}^{N} \cos\left(f^{(j)}(x_j), f^{(k)}(x_k)\right) - \frac{1}{2(N-1)} \sum_{j=k}^{m} \sum_{i=1}^{N} \cos(distri\_f^{(j)}, distri\_f^{(k)})$$
(17)

The model extracts features from each modality branch and maximizes their paired UniFast HGR values in an additive manner. From an information theory perspective, as shown in equation (17), maximizing UniFast HGR is equivalent to extracting the shared information between multiple random variables. This process identifies and leverages the common information content between different patterns or random variables involved.

### 3.4 OPTIMIZATION IN SPEED

To further accelerate the algorithm's computational speed, OptFast HGR was developed as an extension of UniFast HGR, prioritizing efficiency while maintaining reasonable accuracy. The primary improvement in OptFast HGR involves reducing the number of normalization steps, achieving efficiency and computational cost comparable to a dot product operation. This optimization significantly increases computation speed. However, the trade-off for this enhancement is a slight bias introduced in the results. This bias results in correlation values that are marginally shifted due to the reduced normalization steps, highlighting a trade-off between speed and accuracy. While the dot product operation in OptFast HGR provides faster computations, it slightly compromises the precision of the correlation values.

This difference underscores that OptFast HGR, while optimized for speed, may not always align perfectly with the theoretical correlations expected in certain contexts. Nonetheless, the strength of OptFast HGR lies in its ability to process large datasets and models at a significantly faster rate, making it especially suitable for scenarios where computational speed takes precedence over minor variations in accuracy.

The computational process of OptFast HGR algorithm is detailed in Algorithm 2 in A APPENDIX.

## 4 EXPERIMENTS

### 4.1 EXPERIMENTAL SETUP

Experiments were conducted on a range of datasets to assess the performance of the proposed Uni-Fast HGR method. These datasets included multimodal sets with varying features and patterns. The method was implemented using the deep learning framework PyTorch 2.1.1 and Python 3.9.16. All experiments were performed on a 64-bit Ubuntu 20.04 system equipped with dual Intel(R) Xeon(R) Gold 6133 CPUs (2.50 GHz, 40 cores) and dual NVIDIA GeForce RTX3090 GPUs (24 GB memory). The setup also utilized CUDA 12.2 and cuDNN 8.8.

### 4.2 RESULTS AND ANALYSIS

#### 4.2.1 EXECUTION TIME AND FEATURE DIMENSION COMPARISON

The execution times and maximum achievable feature dimensions of various methods, including HGR, Soft-HGR, and UniFast HGR, were compared using the MNIST dataset (LeCun et al., 1998). Following the experimental frameworks of Wang et al. (2019) and Andrew et al. (2013), the left and right halves of each digit image were treated as two distinct patterns. To highlight the efficiency differences introduced by the UniFast HGR, all feature transformations were constrained to a linear form, reducing the maximum correlation of HGR to linear CCA.

As depicted in Figure 2, the execution times for UniFast HGR and OptFast HGR were significantly faster than those of CCA and Deep CCA methods, and also outperformed Soft-HGR. The execution time for the CCA method increased substantially as feature dimensions grew, posing challenges in real-world applications where feature dimensions are typically large. Notably, when the feature dimension exceeded 350, CCA encountered numerical stability issues.

#### 4.2.2 IMAGE CLASSIFICATION

The performance of UniFast HGR was evaluated against several methods, including CCA, Deep CCA, Soft CCA, Soft-HGR, cosine similarity, and dot product, in the context of image classification. Comparative experiments were conducted using a dual-channel deep learning framework for remote sensing data classification, with ResNet 50 as the backbone. Following the same conditions and

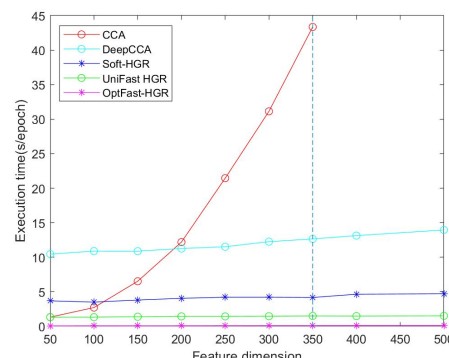

Figure 2: Execution Time and Feature Dimension Comparison on MNIST dataset.

Table 1: Image classification results on the Berlin dataset

| Methods | OA(%) | AA(%) | Kappa(%) | Time(s/epoch) |
|---|---|---|---|---|
| CCA | 70.93 | 64.35 | 58.28 | 2967.52 |
| Deep CCA | 72.74 | 65.08 | 60.23 | 250.51 |
| Soft CCA | 71.54 | 61.14 | 58.33 | 314.93 |
| Dot Product | 75.20 | 66.22 | 62.77 | 23.18 |
| Cosine Similarity | 75.51 | 65.53 | 62.53 | 23.40 |
| Soft-HGR | 65.80 | 64.30 | 52.99 | 25.83 |
| UniFast HGR | 80.75 | 71.53 | 70.44 | 24.53 |
| OptFast-HGR | 80.46 | 71.51 | 70.21 | 23.54 |

preprocessing steps outlined by (Wu et al., 2022) , classification results on the Berlin dataset (Hong et al., 2021; Akpona et al., 2016) are presented in Table 1. The performance was evaluated using three metrics: overall accuracy (OA), average accuracy (AA), and kappa coefficient.

Additionally, experiments were conducted on a dual-channel Vision Transformers framework (Dosovitskiy et al., 2021) for remote sensing data classification. The classification results on the Houston 2018 dataset (Lin et al., 2023) are summarized in Table 2.

UniFast HGR demonstrated competitive performance across all methods, confirming its effectiveness in image classification tasks. Furthermore, OptFast HGR, optimized by reducing the number of normalization steps, achieved computational efficiency comparable to dot product and cosine similarity operations. These results highlight the significant advantages of UniFast HGR and OptFast HGR in classification performance.

### 4.2.3 MULTIMODAL EMOTION RECOGNITION

The performance of UniFast HGR and OptFast HGR was also evaluated in the context of multimodal emotion recognition, with comparisons made to the same set of methods using the IEMOCAP dataset. Comparative experiments were conducted on the MultiEMO model, as proposed by Shi & Huang (2023) . The results of these emotion recognition experiments on the IEMOCAP dataset (Busso et al., 2008) are presented in Table 3. Performance was assessed using the weighted average of the F1 score (W-F1) and accuracy (ACC). Both UniFast HGR and OptFast HGR exhibited strong performance in this task, demonstrating their ability to effectively capture correlations across different modalities within emotion recognition contexts.

### 4.2.4 DISCUSSION

The proposed methods offer several significant advancements for multimodal feature extraction and related applications. First, they provide a more efficient and stable approach for extracting rele-

Table 2: Image classification results on the Houston2018 dataset

| Methods | OA(%) | AA(%) | Kappa(%) | Time(s/epoch) |
|---|---|---|---|---|
| CCA | 88.28 | 92.20 | 84.89 | 1243.23 |
| Deep CCA | 89.82 | 93.92 | 86.89 | 1520.09 |
| Soft CCA | 88.81 | 93.14 | 85.62 | 929.50 |
| Dot Product | 91.59 | 93.85 | 89.13 | 48.89 |
| Cosine Similarity | 92.04 | 94.67 | 89.65 | 49.34 |
| Soft-HGR | 85.86 | 91.01 | 81.91 | 58.03 |
| UniFast HGR | 93.65 | 96.15 | 91.77 | 57.00 |
| OptFast HGR | 93.25 | 95.71 | 91.25 | 52.41 |

Table 3: Experimental results of multimodal emotion recognition on the IEMOCAP dataset

| Methods | W-F1 | ACC | Time(s/epoch) |
|---|---|---|---|
| CCA | 67.51 | 67.41 | 22.62 |
| Deep CCA | 67.82 | 67.78 | 23.72 |
| Soft CCA | 68.57 | 68.58 | 20.94 |
| Dot Product | 69.87 | 70.14 | 19.34 |
| Cosine Similarity | 69.60 | 69.50 | 19.73 |
| Soft-HGR | 71.43 | 71.29 | 21.04 |
| UniFast HGR | 73.57 | 73.66 | 19.56 |
| OptFast HGR | 73.32 | 73.43 | 19.40 |

vant features from multimodal data. The UniFast HGR method reduces computational complexity while improving convergence speed, making it well-suited for large-scale datasets and real-time applications. Second, its capacity to integrate multiple modes increases its flexibility and applicability across various multimodal scenarios, enabling it to handle datasets with diverse patterns. Furthermore, the OptFast HGR approach is optimized by reducing the number of normalization steps, achieving a level of efficiency and computational cost comparable to dot product and cosine similarity operations. Overall, the results indicate that the improved methods not only enhance computational efficiency but also maintain competitive performance in both image classification and multimodal emotion recognition tasks. These attributes position UniFast HGR and OptFast HGR as promising approaches for multimodal feature extraction in a range of applications.

## 5 CONCLUSION AND FUTURE WORK

This paper presented the UniFast HGR framework, which significantly enhances the computation of maximal HGR correlation by incorporating variance constraints and optimizing trace terms. These advancements lead to improved efficiency, scalability, and reduced complexity while maintaining a balanced trade-off between speed and accuracy. As a result, UniFast HGR is particularly well-suited for large-scale neural networks and multimodal applications. The development of OptFast HGR further improves computational speed by reducing the number of normalization steps, achieving efficiency and complexity comparable to that of dot product and cosine similarity operations. However, this approach introduces a slight bias that warrants further investigation. Future work will focus on addressing this bias, potentially through techniques inspired by attention mechanisms, such as those used in Vision Transformers, where proportional adjustments may help mitigate discrepancies between dot product and cosine similarity. Expanding the framework to handle two-dimensional data, such as images, directly within UniFast HGR could also enhance its applicability in deep learning tasks. Additionally, testing the framework on datasets with diverse correlation characteristics, including both positive and negative correlations, will be essential for evaluating its effectiveness and scalability in a broader range of scenarios. In conclusion, UniFast HGR and OptFast HGR offer a scalable and efficient solution for large-scale multimodal learning. However, ongoing refinements will be necessary to maximize their potential in complex real-world applications.

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

## A   APPENDIX: ALGORITHM 2

OptFast HGR, based on UniFast HGR, further accelerates the algorithm's computation speed by reducing the number of normalization steps. The calculation process of the proposed OptFast-HGR algorithm is shown in Algorithm 2 below.

---

**Algorithm 2** OptFast HGR algorithm

---

**Input:** $m \times n$ feature matrix of $\boldsymbol{f}, \boldsymbol{g}$
**Output:** Objective value of OptFast HGR

1. Initialization processing:
   Generation of $t_R$ random matrix $h$ of the same scale as $f$
2. Calculation of HGR deviation:
   $HGR\_\text{bias} = \frac{2}{3t_R} \sum_{i=1}^{t_R} \text{OptFast} - \text{HGR}(h_i, 0)$
   where $0$ is bias
3. Normalization:
   $\mathbf{f} \leftarrow \frac{f}{\|f\|}, \mathbf{g} \leftarrow \frac{g}{\|g\|}$
4. Calculation of the cosine correlation coefficient between $f$ and $g$:
   $corr = \frac{1}{N-1} \sum_{i=1}^{N} \cos(f, g)$
5. Calculation of the distribution matrix:
   $distri\_f = f \cdot f^T, \quad distri\_g = g \cdot g^T$
6. Initialization processing:
   $distri\_f \leftarrow$ The upper triangular part of $distri\_f$ is extracted using the torch.triu function, excluding diagonal elements
   $distri\_g \leftarrow$ The upper triangular part of $distri\_g$ is extracted using the torch.triu function, excluding diagonal elements
7. Calculation of the cosine correlation coefficient between $distri\_f$ and $distri\_g$:
   $tr = \frac{1}{N-1} \sum_{i=1}^{N} \cos(distri\_f, distri\_g) = \frac{1}{N-1} \sum_{i=1}^{N} distri\_f \cdot distri\_g$
8. Calculation of the OptFast HGR objective:
   $\left( \frac{1}{N-1} \sum_{i=1}^{N} \cos(f, g) - \frac{1}{2(N-1)} \sum_{i=1}^{N} \cos(distri\_f, distri\_g) \right) / (1 - HGR\_\text{bias})$

---

## B   DATASET DESCRIPTION

**HSI-SAR Berlin Dataset**: This dataset depicts the city of Berlin and its surrounding regions, including EnMAP HSI images simulated from HyMap HSI data and the corresponding Sentinel-1 SAR data covering the same area. The HSI image is of size $797 \times 220$ pixels, containing 244 spectral bands within the wavelength range of 400-2500 nm. Meanwhile, the SAR data is a dual-polarized SAR, involving four bands (VV-VH). Additionally, a ground truth was generated based on OpenStreetMap data. Training and test sets containing 2820 and 461851 pixels are provided for this dataset, as shown in Table 4.

**Houston 2018 Dataset**: The Houston 2018 dataset, captured by the Hyperspectral Image Analysis Laboratory and the National Center for Airborne Laser Mapping (NCALM) at the University of Houston, was released for the 2018 IEEE GRSS Data Fusion Contest. It covers the University of Houston campus and the neighboring urban area. The dataset includes hyperspectral data with a spectral range of 380–1050 nm across 48 bands and a spatial resolution of 1 meter. Additionally, the LiDAR data is a multispectral image with three bands at 1550 nm, 1064 nm, and 532 nm. This comprehensive dataset represents a challenging urban land-cover and land-use classification task, making it a valuable resource for remote sensing research. Tables 5 shows the number of samples for both training and testing on Houston 2018 dataset.

**IEMOCAP Dataset**: IEMOCAP contains dyadic conversation videos between pairs of ten unique speakers. It includes 7,433 utterances and 151 dialogues. Each utterance is annotated with one of six emotion labels: happiness, sadness, neutral, anger, excitement and frustration. The dataset is divided into separate training and testing sets, and the emotion distribution information of IEMOCAP dataset is shown in Table 6.

Table 4: Berlin dataset with number of training and test samples

| No | Class Name | Training Set | Testing Set | Total Set |
|----|-----------|-------------|-------------|-----------|
| 1 | Forest | 443 | 54511 | 54954 |
| 2 | Residential Area | 423 | 268219 | 268642 |
| 3 | Industrial Area | 499 | 19067 | 19566 |
| 4 | Low Plants | 376 | 58906 | 59282 |
| 5 | Soil | 331 | 17095 | 17426 |
| 6 | Allotment | 280 | 13025 | 13305 |
| 7 | Commercial Area | 298 | 24526 | 24824 |
| 8 | Water | 170 | 6502 | 6672 |
| | Total | 2820 | 461851 | 464671 |
| | Percentage | 0.61% | 99.39% | 100% |

Table 5: Houston2018 dataset with number of training and test samples

| No. | Class Name | Training Set | Testing Set | Total Set |
|-----|-----------|-------------|-------------|-----------|
| 1 | Healthy grass | 1000 | 38196 | 39196 |
| 2 | Stressed grass | 1000 | 129008 | 130008 |
| 3 | Artificial turf | 1000 | 1736 | 2736 |
| 4 | Evergreen trees | 1000 | 53322 | 54322 |
| 5 | Deciduous trees | 1000 | 19172 | 20172 |
| 6 | Bare earth | 1000 | 17064 | 18064 |
| 7 | Water | 500 | 564 | 1064 |
| 8 | Residential buildings | 1000 | 157995 | 158995 |
| 9 | Non-residential buildings | 1000 | 893769 | 894769 |
| 10 | Road | 1000 | 182283 | 183283 |
| 11 | Sidewalks | 1000 | 135035 | 136035 |
| 12 | Crosswalks | 1000 | 5059 | 6059 |
| 13 | Major thoroughfares | 1000 | 184438 | 185438 |
| 14 | Highways | 1000 | 38438 | 39438 |
| 15 | Railways | 1000 | 26748 | 27748 |
| 16 | Paved parking lots | 1000 | 44932 | 45932 |
| 17 | Unpaved parking lots | 250 | 337 | 587 |
| 18 | Cars | 1000 | 25289 | 26289 |
| 19 | Trains | 1000 | 20479 | 21479 |
| 20 | Stadium seats | 1000 | 26296 | 27296 |
| | Total | 18750 | 2000160 | 2018910 |
| | Percentage | 0.9287% | 99.0713% | 100% |

## C  DETAILED EXPERIMENTAL RESULTS

Table 7 presents the detailed comparative experimental results of remote sensing data classification using a dual-channel deep learning framework with ResNet 50 as the backbone on the Berlin dataset. Table 8 displays the results of using a dual-channel visual transformer framework on the Houston 2018 dataset for the same task.

The results demonstrate that the proposed UniFast HGR and OptFast HGR methods consistently outperform traditional Canonical Correlation Analysis (CCA) and similarity-based methods. This suggests that our proposed methods effectively capture complex data patterns and significantly enhance classification performance on both the Berlin HIS-SAR and Houston 2018 HSI LiDAR datasets, irrespective of the framework used (CNN or transformer).

Table 6: IEMOCAP dataset with number of training and test samples

| No | Class Name | Training Set | Testing Set | Total Set |
|----|------------|--------------|-------------|-----------|
| 1 | Happy | 504 | 144 | 648 |
| 2 | Sad | 839 | 245 | 1084 |
| 3 | Neutral | 1324 | 384 | 1708 |
| 4 | Angry | 933 | 170 | 1103 |
| 5 | Excited | 742 | 299 | 1041 |
| 6 | Frustrated | 1468 | 381 | 1849 |
| | Total | 5810 | 1623 | 7433 |
| | Percentage | 78.16% | 21.84% | 100% |

Table 7: Comparison of various methods on the Berlin HIS-SAR dataset(%)

| Class | CCA | Deep CCA | Soft CCA | Dot Product | Cosine Similarity | Soft HGR | UniFast HGR | OptFastR HGR |
|-------|-----|----------|----------|-------------|-------------------|----------|-------------|--------------|
| OA | 70.93 | 71.54 | 72.74 | 75.20 | 75.51 | 65.80 | 80.75 | 80.46 |
| AA | 64.35 | 61.14 | 65.08 | 66.22 | 65.53 | 64.30 | 71.53 | 71.51 |
| Kappa | 58.28 | 58.33 | 60.23 | 62.77 | 62.53 | 52.99 | 70.44 | 70.21 |
| Forest | 81.90 | 87.16 | 64.17 | 76.68 | 79.92 | 67.54 | 87.61 | 82.18 |
| Residential area | 72.81 | 75.59 | 76.38 | 82.57 | 85.63 | 63.87 | 86.85 | 85.10 |
| Industrial area | 23.05 | 53.61 | 76.00 | 48.15 | 49.11 | 64.07 | 40.20 | 62.67 |
| Low plants | 71.44 | 62.68 | 89.08 | 65.08 | 54.31 | 82.05 | 73.70 | 89.23 |
| Soil | 85.97 | 78.01 | 72.10 | 82.53 | 82.88 | 88.16 | 82.42 | 78.63 |
| Allotment | 69.87 | 51.72 | 58.73 | 70.73 | 69.07 | 55.79 | 65.35 | 65.65 |
| Commercial area | 56.76 | 42.81 | 20.40 | 35.88 | 23.77 | 37.97 | 54.30 | 27.61 |
| Water | 52.98 | 37.53 | 63.78 | 68.15 | 79.58 | 54.95 | 81.85 | 81.01 |

Traditional CCA appears less effective at capturing the intricate nonlinear relationships inherent in remote sensing data. Deep CCA exhibits a modest improvement over CCA, suggesting that the integration of deep learning techniques can more effectively grasp these nonlinearities. Both Cosine Similarity and Dot Product perform admirably, highlighting the efficacy of straightforward vector operations for the given datasets. In contrast, Soft HGR underperforms, particularly in Overall Accuracy (OA) metrics, likely due to its propensity to induce substantial alterations in covariance and matrix trajectories, potentially leading to gradient explosions and diminished model efficacy.

The emotion recognition experiments on the IEMOCAP dataset, as detailed in Table 9, indicate that the UniFast HGR and OptFast HGR methods generally excel over conventional CCA and similarity-based approaches. This suggests their enhanced capability for multimodal emotion recognition. UniFast HGR and OptFast HGR demonstrate superior performance across all classifications, showcasing their capacity to effectively capture the nuanced patterns associated with various emotions. Thus, the proposed methods are highly appropriate for emotion recognition tasks and could be applied to other datasets and domains. Future research could integrate these methods with additional modalities like facial expressions and physiological signals to further refine emotion recognition performance.

Table 8: Comparison of various methods on the Houston 2018 HSI-LiDAR dataset(%)

| Class | CCA | Deep CCA | Soft CCA | Dot Product | Cosine Similarity | Soft HGR | UniFast HGR | OptFastR HGR |
|---|---|---|---|---|---|---|---|---|
| OA | 88.28 | 89.82 | 88.81 | 91.59 | 92.04 | 85.86 | 93.65 | 93.25 |
| AA | 92.20 | 93.92 | 93.14 | 93.85 | 94.67 | 91.01 | 96.15 | 95.71 |
| Kappa | 84.89 | 86.89 | 85.62 | 89.13 | 89.65 | 81.91 | 91.77 | 91.25 |
| Healthy grass | 95.62 | 97.84 | 97.97 | 78.15 | 98.24 | 98.76 | 95.18 | 97.66 |
| Stressed grass | 86.77 | 83.27 | 89.16 | 97.58 | 89.66 | 83.84 | 93.57 | 93.27 |
| Artificial turf | 100.00 | 99.83 | 100.00 | 100.00 | 100.00 | 100.00 | 100.00 | 100.00 |
| Evergreen trees | 99.05 | 98.28 | 97.81 | 96.15 | 98.95 | 97.80 | 99.37 | 98.45 |
| Deciduous trees | 96.05 | 95.18 | 95.92 | 94.94 | 97.57 | 96.69 | 98.75 | 98.01 |
| Bare earth | 100.00 | 100.00 | 100.00 | 99.99 | 100.00 | 99.99 | 100.00 | 99.99 |
| Water | 100.00 | 100.00 | 100.00 | 100.00 | 100.00 | 100.00 | 100.00 | 100.00 |
| Residential buildings | 94.02 | 97.90 | 97.42 | 96.88 | 91.92 | 98.49 | 97.04 | 98.20 |
| Non-residential buildings | 94.80 | 94.53 | 93.48 | 95.92 | 97.47 | 91.40 | 98.89 | 96.86 |
| Road | 56.85 | 69.52 | 62.37 | 74.35 | 69.20 | 50.99 | 82.82 | 79.26 |
| Sidewalks | 81.24 | 78.02 | 71.27 | 73.72 | 83.17 | 65.75 | 82.75 | 78.53 |
| Crosswalks | 76.18 | 95.93 | 87.92 | 91.78 | 91.40 | 74.92 | 96.82 | 92.96 |
| Major thoroughfares | 73.24 | 79.62 | 82.78 | 85.45 | 86.32 | 78.80 | 8547 | 87.16 |
| Highways | 98.90 | 95.04 | 96.08 | 97.65 | 99.47 | 96.73 | 98.24 | 99.67 |
| Railways | 99.77 | 99.87 | 99.87 | 99.60 | 99.50 | 99.40 | 99.94 | 99.90 |
| Paved parking lots | 92.95 | 96.88 | 94.18 | 97.46 | 92.83 | 93.98 | 97.02 | 95.53 |
| Unpaved parking lots | 100.00 | 100.00 | 100.00 | 100.00 | 100.00 | 94.07 | 100.00 | 100.00 |
| Cars | 99.13 | 97.41 | 97.17 | 97.45 | 97.65 | 98.53 | 99.16 | 98.70 |
| Trains | 99.95 | 99.41 | 99.57 | 100.00 | 100.00 | 100.00 | 99.99 | 100.00 |
| Stadium seats | 99.57 | 99.94 | 99.83 | 100.00 | 100.00 | 100.00 | 99.98 | 100.00 |

Table 9: Comparison of various methods on the IEMOCAP dataset(%)

| Class | CCA | Deep CCA | Soft CCA | Dot Product | Cosine Similarity | Soft HGR | UniFast HGR | OptFastR HGR |
|---|---|---|---|---|---|---|---|---|
| W-F1 | 67.51 | 67.82 | 68.57 | 69.87 | 69.60 | 71.43 | 73.57 | 73.32 |
| ACC | 67.41 | 67.78 | 68.58 | 70.14 | 69.50 | 71.29 | 73.66 | 73.43 |
| Happy | 50.77 | 49.81 | 46.77 | 50.51 | 53.85 | 54.92 | 66.63 | 59.67 |
| Sad | 79.65 | 81.82 | 79.29 | 81.96 | 81.39 | 81.53 | 84.79 | 85.23 |
| Neutral | 68.11 | 69.58 | 69.59 | 71.24 | 71.89 | 70.84 | 74.30 | 73.00 |
| Angry | 61.98 | 62.53 | 64.60 | 65.90 | 65.82 | 70.32 | 70.46 | 71.04 |
| Excited | 76.70 | 76.56 | 75.00 | 74.48 | 74.91 | 75.00 | 77.14 | 77.09 |
| Frustrated | 60.66 | 59.35 | 65.62 | 67.32 | 63.17 | 69.45 | 71.22 | 70.36 |

## D    REMOTE SENSING SEMANTIC SEGMENTATION

To further evaluate the performance of UniFast HGR and OptFast HGR, we also conducted remote sensing semantic segmentation experiments and compared them with the same set of methods using the Vaihingen dataset.

The ISPRS Vaihingen dataset is a remote sensing image dataset used for 2D semantic segmentation, provided by the International Society for Photogrammetry and Remote Sensing (ISPRS) (Wang et al., 2022). The Vaihingen dataset has a spatial resolution of 9 centimeters and contains 8-bit TIFF files for the near-infrared, red, and green bands, as well as a single band digital surface model (DSM) with height values encoded in 32-bit floating-point numbers. This dataset includes five foreground

Table 10: Comparison of various methods on the Vaihingen dataset

| Class | CCA | Deep CCA | Soft CCA | Dot Product | Cosine Similarity | Soft HGR | UniFast HGR | OptFastR HGR |
|---|---|---|---|---|---|---|---|---|
| OA(%) | 91.15 | 91.39 | 91.41 | 92.61 | 92.56 | 90.10 | 93.01 | 92.95 |
| mF1(%) | 88.12 | 89.45 | 89.50 | 91.14 | 90.61 | 86.46 | 91.35 | 91.23 |
| mIoU(%) | 79.37 | 81.35 | 81.44 | 83.65 | 83.34 | 76.87 | 84.62 | 84.57 |
| Kappa(%) | 88.09 | 88.41 | 88.43 | 90.51 | 89.91 | 86.64 | 91.07 | 90.92 |
| Time (s/epoch) | 1215.31 | 1025.00 | 1136.02 | 1022.78 | 1023.06 | 1034.14 | 1023.82 | 1023.58 |
| Imp. | 91.43 | 92.57 | 92.52 | 94.97 | 93.38 | 91.39 | 93.62 | 93.47 |
| Building | 97.37 | 96.94 | 97.19 | 95.55 | 97.62 | 95.93 | 97.86 | 97.92 |
| Low. | 80.19 | 79.51 | 79.62 | 80.36 | 81.94 | 73.08 | 82.03 | 81.86 |
| Tree | 91.03 | 91.53 | 91.24 | 94.93 | 92.67 | 93.41 | 93.82 | 93.79 |
| Car | 76.94 | 82.94 | 83.76 | 83.41 | 88.53 | 73.86 | 90.15 | 89.95 |

categories, namely Impervious surface (Imp.), Building, Low vegetation (Low.), Tree, Car and one background class (Clutter).

Based on the same model and preprocessing steps outlined by Ma et al. (2024), correlation methods such as UniFast HGR and OptFast HGR were used to fuse multimodal remote sensing data. The experimental results of remote sensing semantic segmentation on the Vaihingen dataset are shown in Table 10. Evaluate performance using overall accuracy (OA), mean F1 score (mF1), and joint mean intersection (mIoU). UniFast HGR and OptFast HGR both demonstrated strong performance in this task, demonstrating their ability to effectively capture correlations between different modalities in high-resolution remote sensing semantic segmentation backgrounds. Figure 3 shows a visualization example of the results obtained using 8 correlation methods. It is evident that when using UniFast HGR and OptFast HGR, complex long-distance semantic information can be more accurately recognized, and precise edges of the recognized object can be obtained, thereby achieving more accurate semantic segmentation of remote sensing imagery.

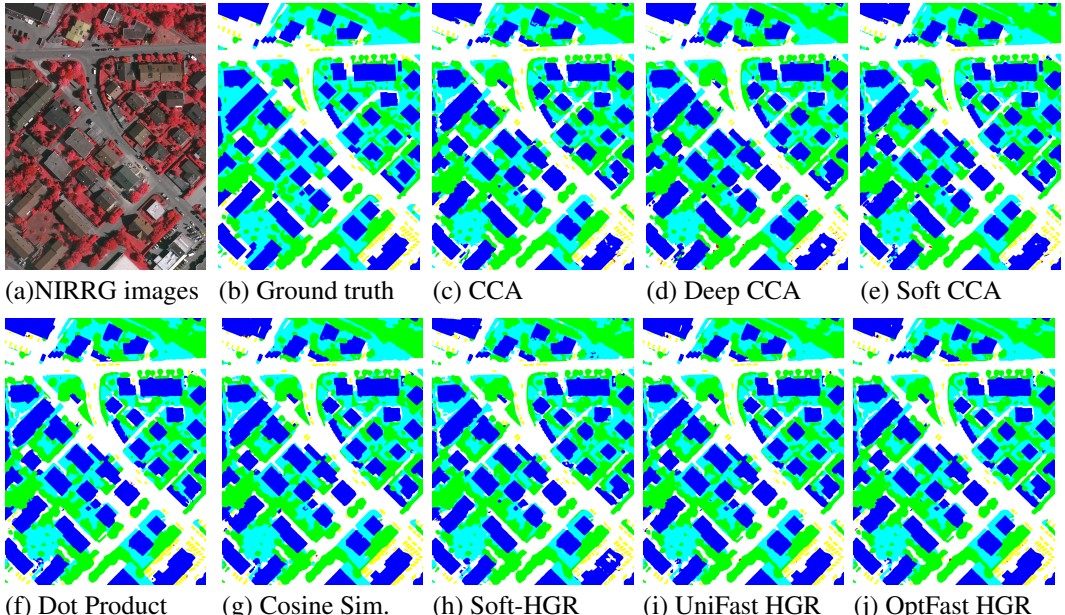

(a)NIRRG images    (b) Ground truth    (c) CCA    (d) Deep CCA    (e) Soft CCA

(f) Dot Product    (g) Cosine Sim.    (h) Soft-HGR    (i) UniFast HGR    (j) OptFast HGR

Figure 3: Experimental images on the Vaihingen test set

Table 11: Execution Time Comparison on the Berlin and Houston2018 datasets (Time(s/epoch))

| Method | ResNet 50 | | Vision Transformer | |
|---|---|---|---|---|
| | Berlin dataset | Houston2018 dataset | Berlin dataset | Houston2018 dataset |
| CCA | 2967.52 | / | 307.82 | 1243.23 |
| Deep CCA | 250.51 | 1158.42 | 379.82 | 1520.09 |
| Soft CCA | 314.93 | 1751.98 | 211.03 | 929.50 |
| Dot Product | 23.18 | 106.05 | 20.85 | 48.89 |
| Cosine Similarity | 23.40 | 106.14 | 20.93 | 49.34 |
| Soft-HGR | 25.83 | 110.53 | 21.62 | 58.03 |
| UniFast HGR | 24.53 | 108.56 | 21.23 | 57.00 |
| OptFast HGR | 23.54 | 106.27 | 21.02 | 52.41 |

# E    COMPUTATIONAL EFFICIENCY

To evaluate the computational efficiency of the proposed UniFast HGR and OptFast HGR, we compared the execution time of remote sensing data classification on the Berlin dataset and the Houston 2018 dataset, using a dual-channel deep learning framework with ResNet-50 as the backbone and a dual-channel visual transformer framework, respectively, as shown in Table 11. The results indicate that CCA, Deep CCA, and Soft CCA had the longest execution times, which were also influenced by the network structure used, whereas UniFast HGR and OptFast HGR were less impacted by these structural complexities.

In order to eliminate the influence of potential confounding factors such as network architecture, we designed an experiment to evaluate the correlation analysis between two randomly generated tensors. The primary objective of this experiment was to compare the execution time of these methods, thereby clearly demonstrating the efficiency advantages of UniFast HGR and OptFast HGR in processing data of varying dimensions.

In this experiment, we compared the computational efficiency of UniFast HGR, OptFast HGR, and other established methods, including CCA, Deep CCA, SoftCCA, and SoftHGR, in calculating the correlation between two randomly generated tensors. The experiment utilized multiple specified dimensional settings and repeated each calculation ten thousand times to ensure result stability. Specifically, the batch sizes (bz) were set to 32, 64, 128, and 256, respectively, and two random tensors, denoted as f and g, with shapes of (bz, dim), were generated using a random function. For each method, the correlation between f and g was computed, and the execution time of each run was recorded. Subsequently, the average execution time for each method across different dimensions was calculated and recorded. The experimental results are shown in Figure 4. The results demonstrate that, as the batch size increases, the execution time gradually grows. UniFast HGR and OptFast HGR consistently exhibit the best performance across different batch sizes, showing significant advantages in computational efficiency. For example, compared to the SoftHGR method, UniFast HGR reduced execution time by approximately 30% to 80%, while OptFast HGR achieved reductions of 50% to 85%.

Moreover, the experimental results indicate that UniFast HGR and OptFast HGR exhibit lower execution times across most dimensions, with their efficiency advantages being particularly pronounced at higher dimensions. These findings suggest that UniFast HGR and OptFast HGR not only effectively capture complex correlations between multimodal data but also demonstrate high computational efficiency, making them well-suited for multimodal data fusion tasks.

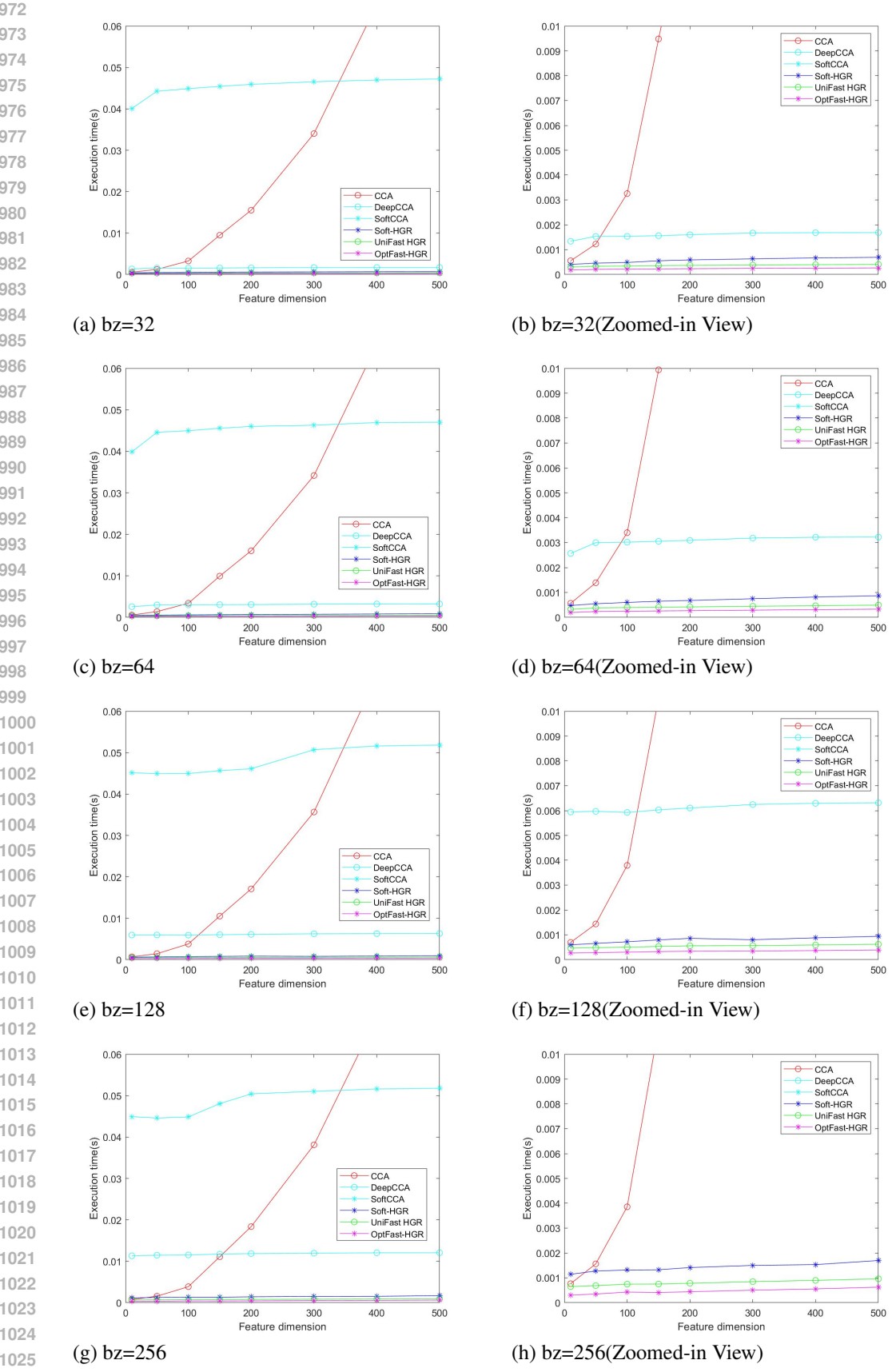

Figure 4: Comparison of Execution Time for Correlation Methods Across Different Batch Sizes and Dimensions Without Interference

