# OpenReview forum: "Fast and Scalable Method for Efficient Multimodal Feature Extraction with Optimized Maximal Correlation"
_ICLR.cc/2025/Conference — ICLR 2025 Conference Withdrawn Submission_

### Official Review · Reviewer_LUtR · 2024-10-31

**Soundness:** 2
**Presentation:** 2
**Contribution:** 2
**Rating:** 6
**Confidence:** 1

**Summary:**

In this paper, a novel approach called the UniFast HGR framework is proposed to optimize the calculation of Hirschfeld-Gebelein-Renyi (HGR) maximum correlation, specifically designed for large-scale neural networks and modal tasks. UniFast HGR introduces variance constraints and optimizes the trace terms, making the approximation to the original HGR more accurate. This method significantly reduces the computational complexity and improves the overall accuracy by using cosine similarity instead of traditional covariance measurement and eliminating the deviation of the main diagonal. These improvements enable UniFast HGR to be highly scalable and perform well in diverse, large-scale modal learning applications. On this basis, the OptFast HGR method further optimizes the performance by reducing the normalization steps, and its efficiency and computational cost are comparable to the dot product and cosine similarity operations. This advancement speeds up computing without sacrificing performance. The results show that UniFast HGR effectively balances efficiency and precision, making it a reliable solution to modern deep learning challenges.

**Strengths:**

1. The proposed method is theorically sounded.

**Weaknesses:**

1. It is difficult to understand this paper, especially for those who are not familiar with the Hirschfeld-Gebelein-R´enyi (HGR) maximal correlation. I have tried my best to understand the proposed method. However, I was trapped by the definition of Eq.(1).  In Eq.(1), the definition of f and g is unclear. Does each f_i/g_i represent a feature extractor? If so, what do 1/k[f], E[g], and Cov(f)/Cov(g) mean?
2. It is also difficult to clearly understand which limitations the proposed UniFast HGR can alleviate. Because the authors have talked about substantial computational burdens, weakly correlated modalities, missing labels, incomplete modalities, and data augmentation. If all these limitations can be solved by the proposed UniFast HGR, it is better to verify them in the experiments.
3. Does UniFast HGR can be applied to the current multi-modal large fundation models, such as CLIP?

**Questions:**

Please see the weakness.

---

### Official Review · Reviewer_soHm · 2024-11-01

**Soundness:** 1
**Presentation:** 1
**Contribution:** 1
**Rating:** 1
**Confidence:** 4

**Summary:**

While Hirschfeld-Gebelein-Re ́nyi (HGR) maximal correlation can be used for learning multi-modal representations, it hasn't gained popularity in the current era of large models due to its computational complexity. This paper introduces an approximation of  HGR maximal correlation that is faster to compute. This is achieved by replacing traditional covariance-based measures with cosine similarity and eliminating bias from the main diagonal. Experiments were conducted on remote sensing and emotion classification datasets.

**Strengths:**

The paper does a good job in presenting the problem and background work in the introduction section.

**Weaknesses:**

The writing of the paper needs significant improvement. The presentation of the proposed methodology is difficult to follow. I tried quite a bit to understand the paper, but there are several errors in Sec.3 that raise questions about the correctness of the content.

Eq(1) - It should be ``E[f] = 0`` instead of ``1/k[f] = 0`` under sup.

Eq(3) and Eq (4) - There is a summation over ``i``, but the subscript ``i`` is missing for ``x`` and ``y``.

Eq(5) - ``f(x)`` and ``g(y)`` are ``k``-dimensional vectors. So, ``Sqrt(Var[f(x)])`` and ``Sqrt(Var[g(y)])`` do not make sense. Variance is valid for a single random variable. For a vector, we get a covariance matrix. ``f(x)`` and ``g(y)`` in this equation should have an extra subscript ``k`` which denotes each individual dimension of the feature representation, and there should be a summation over this ``k``.

Lines 228-233: Authors use the terms similarity and correlation in a confusing manner in these lines: Line 228 says ``correlation between similarity matrices``, line 230 says ``similarity between correlation matrices``, line 233 says ``similarity between similarity matrices``. According to the original Soft-HGR formulation the trace term is actually an ``inner product (element-wise multiplication followed by summation) of two covariance matrices``.

Eq (6) - The definition of the trace term is wrong (according to Soft-HGR formulation). It is ``trace(cov(f)cov(y))`` not ``trace(cov(f,y))``.

Figure 1: I do not understand what ``correlation between similarity matrices of two modalities`` mean. What does ``similarity matrix`` of a modality mean? What does ``correlation between two matrices`` mean?

Eq (9) - Based on my understanding, this is wrong.
a) It should be ``trace(cov(f)cov(y))`` not ``trace(cov(f,y))``.
b) The indices ``i`` and ``j`` denote embedding dimensions. So they should go from ``1`` to ``K``, where ``K`` is the number of dimensions. Instead, the summation currently runs from ``1`` to ``N``, where ``N`` is the number of samples.
c) I do not understand what ``E[cov f_i]`` and ``E[cov g_j]`` mean? ``cov f_i`` and ``cov f_j`` are not defined anywhere.
d) I don't understand why there is a normalization factor of ``(N-1)`` in the denominator.
If we expand the trace term it should be
``trace(cov(f)cov(g)) = \sum_{i=1}^K\sum_{j=1}^K f_{ij}g_{ij}``

Eq(10) - Based on my understanding, this is wrong. It has errors similar to 9(b), 9(c), 9(d) pointed above.

Lines 291-292: These lines say "If all components of a random vector are independent, the square of the vector’s modulus will equal the sum of the variances of each component". This is not correct. It should be "the expected value of the the square of the vector’s modulus". The words "expected value" are crucial for correctness.

Eq (12) is incorrect:
a) It should be ``x_i`` instead of ``X`` and ``y_i`` instead of ``Y``.
b) Even if each individual dimension of ``f`` and ``g`` is assumed to have zero mean and unit variance, it does not guarantee that individual sample vectors ``f(x_i)`` and ``g(y_i)`` will have unit norm.

Eq (13): I do not understand what ``cos f_i`` means. I couldn't follow what is the basis for replacing covariance between two random variables with  cosine similarity between random vectors.

**Questions:**

The paper needs significant rewriting. Based on my understanding, I am not convinced about the technical correctness of the paper. See my comments in "weaknesses" section.

---

### Official Review · Reviewer_9sBw · 2024-11-04

**Soundness:** 2
**Presentation:** 2
**Contribution:** 2
**Rating:** 3
**Confidence:** 5

**Summary:**

For large-scale multimodal applications based on neural networks, the paper proposes a framework to approximate Hirschfeld-Gebelein-Renyi (HGR) more accurately with less computation. Cosine similarity and bias elimination are used instead of the traditional covariance-based measure.  In addition, reducing the normalization steps leads to a lower computational load.

**Strengths:**

The paper's readability is generally good. It provides a nice background behind the development of HGR and clearly states that its goal is to make Soft-HGR more computationally feasible. Experimental results on MNIST, Houston2018, IEMOCAP compared to CCA, Deep CCA, Soft CCA, Soft-HGR, dot product, and cosine similarity show UniFast HGR and OptFast HGR to be better.

**Weaknesses:**

1. Lack of Comprehensive Solution for Large-Scale Multimodal Representation:
While the paper aims to provide a scalable method for efficient multimodal feature extraction, it falls short of presenting a comprehensive solution for large-scale multimodal representation. Although the HGR (Hirschfeld-Gebelein-Rényi) correlation measure might be a reasonable mathematical formulation, other multimodal approaches, such as CLIP, have proven efficient and effective for large-scale applications. CLIP, which utilizes a dot-product-based representation, offers both scalability and effectiveness across vast datasets. To strengthen the paper, the authors could conduct specific comparisons or experiments with methods like CLIP to highlight the advantages or limitations of the proposed approach. Including examples of large-scale multimodal applications where the proposed method might face challenges would further clarify its suitability and potential limitations in broader contexts.


2. Dataset Size and Evaluation on Large-Scale Datasets:
The dataset used in this study is not large-scale by current standards. Demonstrating the method’s effectiveness and scalability on significantly larger datasets—preferably 100 times larger than those currently used—would provide a more convincing evaluation of its applicability for large-scale multimodal data. Additionally, since the method is intended to be a general-purpose multimodal representation, testing it across various combinations of multimodal datasets would offer valuable insights into its scalability and robustness. Specific suggestions for appropriate larger datasets or multimodal combinations would benefit the evaluation.
Please consider the dataset in the following papers:
Alec Radford et al. Learning Transferable Visual Models From Natural Language Supervision CVPR 2021.
Yi Wang et al. InternVid: A Large-scale Video-Text Dataset for Multimodal Understanding and Generation CVPR 2023

There are more at https://github.com/drmuskangarg/Multimodal-datasets.

3. Lack of Discriminative Power in HGR:
HGR, based on correlation, appears to lack a discriminative element often necessary for effective multimodal representation learning. The absence of discriminative capabilities may limit the model’s ability to distinguish between different categories or classes within multimodal data, which is essential for many applications. Providing evidence or demonstrations of HGR’s discriminative limitations or suggesting ways to improve the model's discriminative properties, perhaps by incorporating discriminative loss functions or alternate approaches, would strengthen the paper’s technical contribution.

4. Undefined Notation for Covariances
The notations cov(fi) and cov(gi) are not defined, confusing readers who are attempting to follow the mathematical formulation. Adding clear definitions for these terms in the relevant sections and explanations of their role within the HGR-based formulation would improve the paper's readability and accuracy.

The paper introduces a scalable feature extraction method for multimodal data but lacks a comprehensive evaluation for large-scale applications, especially compared to established approaches like CLIP. Addressing dataset scale, incorporating discriminative elements, and clarifying notation would improve the study’s rigor and relevance.

**Questions:**

See above.

---

### Note · Authors · 2024-11-14

I have read and agree with the venue's withdrawal policy on behalf of myself and my co-authors.